

# Blade Hardness Gauge: Snow Hardness Measuring and Analysis Techniques

Peter K.A. Barsevskis[1,2,3], Mark J. Paetkau[1]

[1]Thompson Rivers University, Kamloops, V2C 0C8, Canada
[2]Kicking Horse Mountain Resort, Golden, V0A 1H0, Canada
[3]Brucejack Mountain Safety, Kitimat-Stikine A, V0T 1W0, Canada

*Correspondence to*: Peter Barsevskis (p.barsevskis@me.com)

**Abstract.** The blade hardness gauge (BHG) is a promising technology for avalanche forecasters, technicians, and researchers.
Designed and produced by Fraser Instruments Ltd., the BHG resembles and is based on the thin-blade tool introduced by Borstad and McClung in 2011. The BHG was designed to quantitatively measure snow hardness without the known biases of the hand hardness test. Research was carried out in the Canadian mountains of British Columbia and Alberta during the 2020-21 and 2021-22 winter seasons to test the reliability and integrity of the BHG. Side by side snow hardness profile comparison amongst avalanche practitioners shows that the BHG is more consistent for measuring snow hardness than the hand hardness
test. A blade hardness to hand hardness comparative scale was developed to utilize the BHG as a teaching tool for the hand hardness test. This paper proposes refinements to standard data collection methods and techniques including the insertion rate and orientation of the thin-blade into the snowpack. These recommendations aim to increase consistency amongst users and highlight applications for avalanche practitioners to use in the field.

## 1 Introduction

Avalanche forecasters interpret current weather and snowpack profiles to predict the outcome of future avalanche trends. Snow profile observations can be taken throughout the entirety of the snowpack to gain knowledge with respect to the snow depth, snow layers, snow hardness, grain form, grain size, liquid water content, and snow density.

A predictive measurement used in avalanche forecasting is snow hardness, which is a measure of the snow's resistance to penetration by an object (Fierz et al., 2009). The resistance is the combination of snow grain bonds and structures, bending,
rupturing, and compacting along with the friction between the snow and the penetrating object (Borstad & McClung, 2011). The current standard for measuring snow hardness in Canada outlined by the Canadian Avalanche Association is the hand hardness test (Canadian Avalanche Association, 2016). This paper focuses on methods and characteristics of measuring snow hardness with the blade hardness gauge (BHG).

The BHG is a promising technology to quantitatively measure snow hardness without the shortcomings of the hand hardness
test. With no standard method of quantitatively measuring snow hardness this research aims to test the reliability and integrity



of the BHG in relation to measuring snow hardness. To test this the research was broken into separate components including insertion rate, gauge replication, orientation of the gauge, and comparison between blade hardness measurements with hand hardness measurements.

The BHGs used in this study were the third and latest model of the BHG produced by Fraser Instruments Ltd. (Fig. 1). The

blade is stainless steel, 0.6 mm thick, width of 10.0 cm, has a force range of 0 – 50 N, and is precise to 0.05 N. The BHG measures and displays the peak resistance hardness as the blade is inserted into the snowpack.

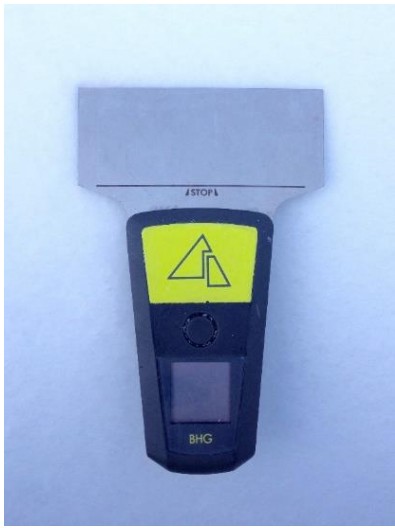

**Figure 1: Blade Hardness Gauge**

The BHG is based off the thin-blade tool introduced by Borstad and McClung (Borstad & McClung, 2011). The thin-blade

device allows the avalanche practitioner to measure the hardness of thin weak layers over time (Pogue & McClung, 2016). A Parks Canada study comparing the use of the BHG and the hand hardness test concluded that: the ± indexes used have no meaning, that 'fist' and '4 fingers' hardness are basically the same, and that further testing is needed for operator hand hardness bias (Pogue et al., 2018).

## 2 Snow Hardness

### 2.1 Snow Hardness History

Snow hardness has been described from as early as 1885 by Heim, a pioneer of glaciology, who described the changes in the hardness from new snow to glacier ice. In 1930, Welzenbach differentiated two categories for hardness, packed snow, and pressed snow. A snow profile technique designed by Paulcke, in 1938, resulted in the first iteration of the hand hardness test. This technique involved describing the ease of penetration of a finger into different stratigraphic snow layers (Pielmeier &

Schneebeli, 2003a).



The first mechanical measurements of snow hardness were taken using the Swiss rammsonde, a metal probe driven into the snow by the observer dropping specified weights on the probe, in 1936 (Haefeli, 1954; Höller & Fromm, 2010). Although capable of measuring snow hardness, it is unable to detect thin weak layers associated with slab avalanches (Schneebeli & Johnson, 1998). The snow resistograph was invented utilizing a probe with a horizontal blade on the end; when the probe gets to the bottom of the snowpack it is turned 90 degrees and withdrawn upwards with the resistance of the snow on the thin part of the blade being recorded (Bradley, 1966). An advantage of the resistograph was that it outputted a graphical hardness profile but had a high cost and high mass (Pielmeier & Schneebeli, 2003a). The snow resistograph showed that varying the rate of withdrawing the blades through the snow varied the resistance measurements with a rate of 10 cm/s being the optimum withdrawing rate for consistency (Bradley, 1966). Further work was done making a digital snow resistograph (Brown & Birkeland, 1990; Dowd & Brown, 1986) but it was ultimately unable to detect thin weak layers and lacked durability (Schneebeli & Johnson, 1998). The electric cone penetrometer (Schaap & Föhn, 1987) tried filling this void of reliably measuring snow hardness but also was unable to detect thin weak layers (Schneebeli & Johnson, 1998). The SnowMicroPen (SMP) is a motor-controlled, high spatial resolution penetrometer (Schneebeli & Johnson, 1998) utilizing a five mm cone diameter, compared to the Swiss rammsondee's 40 mm cone diameter. With the high resolution and smaller cone diameter the SMP can accurately detect the presence of weak layers (Johnson & Schneebeli, 1999; Pielmeier & Schneebeli, 2003b; Schneebeli & Johnson, 1998). Although the SMP can detect weak layers in the snowpack it is not commonly used amongst avalanche practitioners due to its size, weight, and high costs (Lutz & Marshall, 2014).

In 2012 a group of MIT graduates started the company AvaTech to produce a rapid snow penetrometer (Christian et al., 2014). During the winter of 2013-14 the AvaTech SP1 was tested by the manufacturers and independent researchers. The SP1 was designed to record hardness profiles and additional information such as depth of snowpack. Lutz and Marshall (2014) reported the SP1 recorded consistent hardness values, but needed improvements with respect to snowpack depth, data from the upper snowpack and detecting the presence of buried weak layers. In 2016, AvaTech released the SP2 as an improvement to the SP1 (Pielmeier & van Herwijnen, 2016). Between January and March 2016, Pielmeier and van Herwijnen carried out side by side hardness profiles with the SP2, SMP and Swiss rammsonde. It was found that the SMP was still superior to the SP2 for accurate positioning of hardness measurements with respect to snowpack depth and the presence of buried weak layers (Pielmeier & van Herwijnen, 2016).

The hand hardness test has the operator exert 10-15 N of force using physical objects of decreasing surface area (fist (F), 4 fingers (4F), 1 finger (1F), pencil (P), and knife (K)) into the snowpack. This standard has been set by "The International Classification for Seasonal Snow on the Ground" (Fierz et al., 2009). Furthermore, the Canadian Avalanche Association (CAA) has operators add + and - indicators to illustrate variations in hardness (Canadian Avalanche Association, 2016). This test has shortcomings in accuracy: bias amongst users, failure to consistently apply 10-15 N, misusing ± as a classification, and varying size of hand (Pogue et al., 2018).





## 2.2 Thin-blade Hardness Measurements

Fukue (1977) utilized a thin-blade penetration technique for measuring snow characteristics with four goals in mind: (1)
involved a simple technique, (2) minimized effect of speed/penetration rate, (3) minimized densification of snow during
testing, and (4) minimized the change of snow grain bonding during testing. Fukue mounted the thin-blade, 12 mm wide and
0.6 mm thick, onto an actuator that mechanically inserted the blade into the snow. The testing found a ductile to brittle transition
at a penetration rate of 0.25 mm/s with the snow exhibiting ductile properties below 0.25 mm/s and the brittle properties above
0.25 mm/s. When the snow is exhibiting brittle properties, it can be assumed that the thin-blade penetration was measuring
snow characteristics without the change of the initial properties such as density (Fukue, 1977).

In 2011 Borstad and McClung introduced the blade hardness gauge (BHG) to measure snow hardness in a quantitative manner
(Borstad & McClung, 2011). The BHG utilizes a thin-blade (0.6 mm thick), which is the average sized snow grain of alpine
snow and is used to minimize snow compaction. The first iteration had an operating temperature range of -1 to 49 °C and a
load cell capacity of 250 N with a resolution of 0.1 N (Borstad & McClung, 2011). In 2013, Buhler custom built a BHG to
measure the hardness of snow crust layers. It utilized the same size blade as the first iteration designed by Borstad and
McClung. The BHG was able to track the hardness of the snow crust layers over time and found a correlation between hardness
and snow density (Buhler, 2013).

Further iterations of the BHG were produced by Fraser Pogue. In 2016 his first iteration of the BHG utilized 3D printing to
build the enclosure to attach the thin-blade (0.6mm thick, 10 cm wide) to the components of the force sensor. This enclosure
was made to be small, lightweight, and waterproof. The force sensor had a range of 0.05-49 N, was accurate to 0.1% and an
operating temperature range of -20 to 40 °C. The thin-blade allows the avalanche practitioner to measure the hardness of thin
weak layers over time (Pogue & McClung, 2016). During the winter of 2016-17 Parks Canada staff in Glacier National Park
used the BHG in 27 snow profiles by 8 different operators. The BHG used in this study was Pogue's second iteration of the
gauge. This iteration of the BHG was designed to be friendlier for operational use. The Parks Canada study concluded that:
the ± indexes used have no meaning, that fist and 4 fingers hardness is basically the same, and that further testing is needed
for operator hand hardness bias (Pogue et al., 2018).

## 2.3 Calibration of the Hand Hardness Test

In 1950, de Quervain introduced the hand hardness test with five classes. These classes were correlated to the Swiss
rammsonde and relative snow hardness (de Quervain, 1950). The Commission of Snow and Ice (ICSI) of the International
Association of Hydrology (IASH) published the first issue of "The International Classification for Snow (with special reference
to snow on the ground)" in 1954. The commission describes snow hardness as the correlation between the measurements from
a particular hardness instrument and the combination of the compressive yield strength, tensile strength, and shear strength at
zero normal stress of snow (Schaefer et al., 1954).



The United Nations Education, Scientific and Cultural Organization (UNESCO), IASH, and World Meteorological
Organization (WMO) published "Seasonal Snow Cover" in 1970 to update the 1954 international classification of snow. This
updated version related snow hardness to the Swiss rammsonde and the hand hardness test. The standard penetration force for
the hand hardness test was set at a force of 50 N (International Association of Scientific Hydrology et al., 1970).

The International Commission on Snow and Ice (ICSI) of the International Association of Scientific Hydrology published the
third issue of "The International Classification for Seasonal Snow on the Ground" in 1990. Ice was added as a sixth class to
the hardness categories with no correlation to the hand hardness scale, Swiss rammsonde, or magnitude of strength. The
standard penetration force for the hand hardness test was maintained at a force of 50 N (Colbeck et al., 1990).

The latest edition of "The International Classification for Seasonal Snow on the Ground" was published in 2009 by the
International Hydrological Programme of the UNESCO. In 2009 the standard penetration force for the hand hardness test was
set at a force of 10-15 N, which was the standard in North America (McClung & Schaerer, 2006).

During the initial prototyping of the SnowMicroPen (SMP), a motor controlled high spatial resolution penetrometer,
Schneebeli and Johnson compared SMP hardness profiles with the hand hardness test. Comparing the two methods illustrated
that layers of snow with high variability of snow hardness determined by the SMP are difficult to judge by the hand hardness
test (Schneebeli & Johnson, 1998). During the 2000/01 and 2001/02 winter seasons, Pielmeier and Johnson, carried out seven
snow profiles recording adjacent hardness profiles with the hand hardness test, Swiss rammsonde and the SMP. The SMP
proved to yield the highest spatial and force resolution of the three methods. Snow hardness with gradual changes at layer
boundaries were impossible to accurately measure with the hand hardness test but were possible with the SMP (Pielmeier &
Schneebeli, 2003b). During the 2007/08 and 2008/09 winter seasons, Höller and Fromm (2010) carried out a few tens of snow
profiles recording adjacent hardness profiles with the hand hardness test, a digital force gauge and the SMP. They found that
for each hand hardness index there was variability in the measured values of hardness by the SMP, and the digital force gauge.
The variability of hardness measurements within a given hand hardness index illustrates the biases of the hand hardness test.
Utilizing a digital force gauge will show the variations of hardness in a specific snow layer.

Borstad and McClung compared the first iteration of the BHG with the hand hardness test. By comparing 520 BHG
measurements with their respective hand hardness measurements resulted in considerable overlap between blade hardness and
hand hardness classes (Borstad & McClung, 2011). The first iteration of the BHG was unable to measure soft snow with a
hand hardness softer than 4F-. The Parks Canada study with a more current iteration of the BHG during the 2016-17 winter
compared 685 BHG measurements with their respective hand hardness measurements. This study resulted in considerable
overlap between blade hardness and hand hardness classes (Pogue et al., 2018).

Ever since the hand hardness test was first introduced by de Quervain in 1950 it has been calibrated against a variety of
hardness instruments including, the Swiss rammsonde, the SMP, various types of digital force gauges and the BHG. The hand
hardness test is subjective based on the person using it, varying force of insertion and varying size of hand. Using technologies
such as the BHG, the human biases of the hand hardness test may be eliminated resulting in the ability to measure snow
hardness over time with multiple operators.





## 3 Objectives

Utilizing the BHG to quantitatively measure the hardness of snow this research explored the following objectives:

- Determine if there is a difference in recorded BHG measurements between fast (≈ 10 cm/s) and slow (≈ 1-3 cm/s) insertion rates into the snowpack.
- Determine if there is a difference in recorded BHG measurements depending on the orientation of the BHG into the snowpack.
- Find a correlation between BHG measurements and the hand hardness test.
- Test the replication of the hand hardness test versus the BHG amongst avalanche technicians.

## 4 Methods

This study occurred over the winter seasons of 2020/21 and 2021/22. The primary field sites were situated within the Kicking Horse Mountain Resort tenure and the surrounding backcountry in Golden, BC, Canada. Additional field data was gathered in the Canadian Rockies, Rogers Pass, Big White Ski Area, and Whistler Blackcomb. The snow profiles carried out in the 160 designated study sites adhered to the observation and recording guidelines set forth by the Canadian Avalanche Association (Canadian Avalanche Association, 2016). Further observations were made using the BHG.

### 4.1 Insertion Rate

The BHG was inserted into the snowpack with a fast (≈ 10 cm/s) insertion rate while maintaining a slope parallel angle. Subsequently, the same BHG was inserted into the snowpack at the same depth of snow with a slow (≈ 1-3 cm/s) insertion rate 165 at the same angle as the prior measurement (parallel with the slope). The insertions were spaced roughly two cm apart. Figure 2 illustrates the spacing of the BHG measurements that were taken.

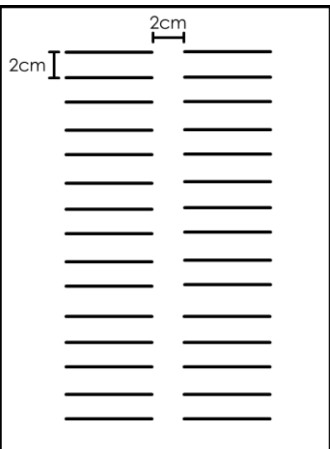

**Figure 2: Schematic of the spacing of blade hardness measurements in relation to the insertion rate objective.**



To determine the consistency of fast versus slow measurements, a second experiment was completed. In this experiment,
layers of homogenous snow greater than 10 cm in height were utilized to take trials of 10 fast versus 10 slow measurements.
The BHG was placed into the snow perpendicular to the slope angle to reduce spatial variability of the snowpack in relation
to snow layering. This procedure was carried out in multiple layers of snow differing in snow hardness. Figure 3 illustrates
the spacing of BHG measurements that were taken.

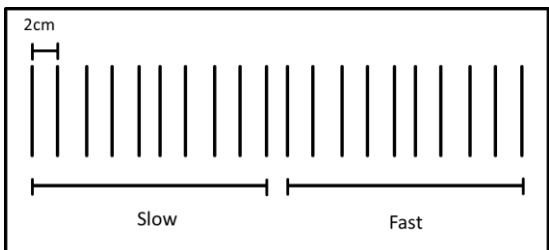

**Figure 3: Schematic of the spacing of blade hardness measurements in relation to the consistency of insertion rate objective.**

In both of the above experiments, the velocity of insertion rates was standardized through timer and ruler-based measurements,
where velocity is the measure of the distance covered in a given amount of time. These calibrations were conducted inside by
the researcher before venturing into the field. During fieldwork, the researcher subjectively assessed the insertion rates.

**4.2 Orientation**

Homogenous snowpack layers with a height of 10 cm or more (determined through a combination of visual and physical
methods in excavated snow profiles) were subject to a comparison between six horizontal BHG measurements (blade aligned
parallel to the slope) and one vertical BHG measurement (blade aligned perpendicular to the slope). For the horizontal
measurements intervals of two cm were employed vertically resulting in a cumulative vertical height of 10 cm. When using
the BHG in a vertical orientation, a measurement was obtained over the entire 10 cm vertical distance. Both horizontal and
vertical measurements were spaced roughly two cm apart and their corresponding values were documented. Figure 4 illustrates
the spacing of the BHG measurements that were taken.

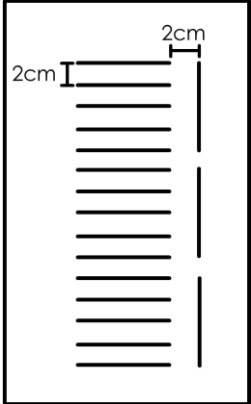

**Figure 4: Schematic of the spacing of blade hardness measurements in relation to the orientation of the blade hardness gauge.**





### 4.3 Hand hardness

Blade hardness measurements were conducted in conjunction with the respective hand hardness profiles, aiming to quantitatively gauge the hand hardness scale. An avalanche technician recorded hand hardness for each layer. The researcher, on the other hand, performed BHG measurements at approximately two cm intervals within the layers to ensure uniform BHG measurements. The insertion rate for all BHG measurements was maintained at approximately 10 cm/s, with the researcher subjectively assessing the rate.

To assess the repeatability of both the hand hardness test and the BHG, avalanche technicians sequentially executed hand hardness and BHG measurements. Each technician was unaware of the measurements previously taken by others to ensure independence. The technicians were instructed to carry out BHG measurements with a consistent, rapid insertion rate of approximately 10 cm/s, every two cm within the layers. These sets of measurements were obtained from the same snow profile, with minimal time gaps between technicians to mitigate weather-related influences, and minimal spacing to mitigate spatial

variability effects in the snowpack.

### 4.4 Statistical Analysis

During the field research the data was written down into "Rite in the Rain" notebooks. The data was then transferred to Microsoft Excel spreadsheets. Statistical analysis was carried out in Microsoft Excel or Minitab statistical software. Blade hardness measurements and density measurements are ratio and continuous data. Hand hardness measurements and extended

column test results are ordinal and continuous data. All data collected were first tested for normality with either the Ryan-Joiner test (n < 25) or the Kolmogorov-Smirnov test (n > 25).

For paired tests (insertion rate and gauge orientation) the Wilcoxon signed-rank test was used to test if there is a statistically significant difference from zero in the median of the distribution of differences of the paired measurements. To compare BHG measurements with respect to their corresponding hand hardness scale, the Kruskal-Wallis test and the Mann-Whitney test will

be utilized to compare the respective medians. The Kruskal-Wallis test will be used to compare three or more hand hardness levels (such as 4F-, 4F and 4F+) while the Mann-Whitney test will be used when comparing two hand hardness levels (such as F and 4F).

To evaluate the hand hardness reproducibility, direct observations of each avalanche technician were compared to one another. To evaluate the BHG reproducibility, a minimum of three BHG measurements were needed by each avalanche technician per

snow layer. The BHG measurements were then compared using either the two-sample t-Test (parametric) or the Mann-Whitney test (non-parametric) depending on the distribution of the data. For both the two-sample t-Test and the Mann-Whitney Test a significance threshold of $p \leq 0.05$ was used. If the comparison resulted in $p > 0.05$ the data supports that the BHG measurements are in agreement of each other within current measurement precision.



## 5 Results

### 5.1 Insertion Rate

Pairs of BHG measurements were taken to test if there is a difference in BHG measurements with respect to fast (≈ 10 cm/s) and slow (≈ 1-3 cm/s) insertion rates. A total of 136 in situ pairs were taken in snow profiles consisting of dry snow ranging in blade hardness 0.1 N to 36.2 N (Table 1). Being paired and nonparametric, the Wilcoxon signed-rank test was used to compare the distribution of differences between the fast and slow insertion rates (WS = 1938.00, $p < 0.01$). The data supports

that there is a statistically significant difference between the median of the distribution of differences between the fast and slow insertion rates (Fig. 5).

| Insertion Rate | Average (N) | Standard Deviation (N) | Standard Error (N) |
|---|---|---|---|
| Fast | 6.47 | 7.54 | 0.65 |
| Slow | 8.01 | 9.00 | 0.77 |
| Difference | -1.54 | 2.69 | 0.23 |

**Table 1: Descriptive statistics comparing fast and slow insertion rates with the BHG (Difference = Fast – Slow).**

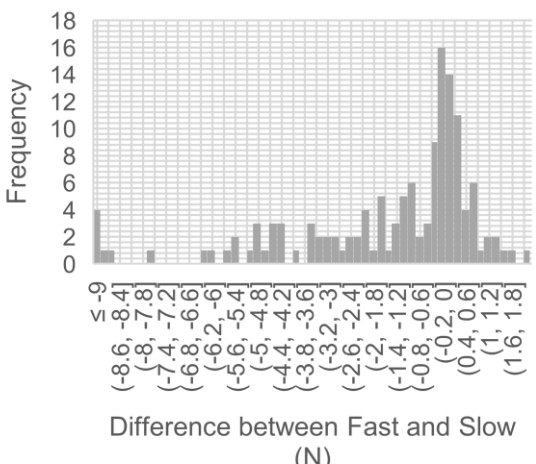

**Figure 5: Histogram of the distribution of differences between the fast and slow insertion rates of the BHG into dry snow, n=136.**

To test the consistency of the insertion rates, trials of 10 fast versus 10 slow measurements were taken in layers of homogenous snow greater than 10 cm in height. This procedure was carried out in multiple layers of snow differing in snow hardness resulting in 11 trials of 10 fast versus 10 slow BHG measurements.

Comparing the standard deviations of the 11 trials (Fig. 6) illustrates that using a fast insertion rate results in more consistent measurements without a high scatter of data. Using a slow insertion rate results in a higher scatter of data in each snow layer.



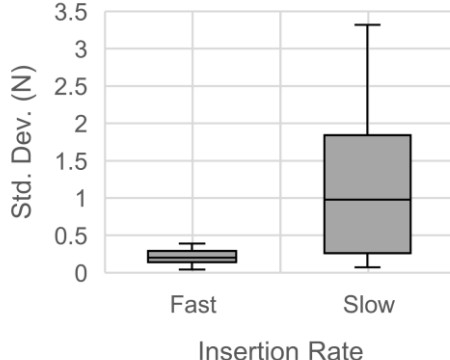


**Figure 6: Box and whisker plot grouping the standard deviations of the 11 fast versus slow insertion rate trials.**

Being paired and nonparametric, the Wilcoxon signed-rank test was used to compare the distribution of differences between the fast and slow standard deviations (WS = 1.50, p < 0.01). The data supports there is a statistically significant difference from zero in the median of the distribution of differences between the fast and slow insertion rates. With the fast insertion rate

consistently having a smaller standard deviation.

**5.2 Orientation**

186 vertical (slope perpendicular) BHG measurements (height of 10 cm) were compared with 186 mean horizontal (slope parallel) measurements (mean of six horizontal measurements spaced two cm apart for total height of 10 cm) in homogenous layers of dry snow greater than 10 cm in height (Table 2). Being paired and nonparametric, the Wilcoxon signed-rank test was

used to compare the two gauges (WS = 6206.00, p < 0.01). The data supports that there is a statistically significant difference from zero in the median of the distribution of differences between the two BHGs.

| Orientation | Average (N) | Standard Deviation (N) | Standard Error (N) |
|---|---|---|---|
| Vertical | 4.90 | 5.61 | 0.41 |
| Mean Horizontal | 5.19 | 6.01 | 0.44 |
| Difference | -0.29 | 1.19 | 0.09 |

**Table 2: Descriptive statistics comparing orientation of the BHG (Difference = Vertical – Mean Horizontal).**

**5.3 Hand Hardness**

During the 2020/21 and 2021/22 winter field seasons a total of 68 hand hardness profiles by 33 different avalanche technicians

were taken with corresponding BHG measurements. The avalanche technicians classified the hand hardness test with the five





hand hardness indices (F, 4F, 1F, P and K) and with the ± indices. A total of 4229 BHG measurements were compared with the hand hardness indices (Table 3 and Fig. 7).

| Hand Hardness | Count | Mean (N) | Median (N) | Standard Deviation (N) | Standard Error (N) |
|---|---|---|---|---|---|
| F- | 25 | 0.16 | 0.14 | 0.14 | 0.03 |
| F | 240 | 0.28 | 0.25 | 0.22 | 0.01 |
| F+ | 64 | 0.36 | 0.30 | 0.30 | 0.04 |
| F (total) | 329 | 0.28 | 0.24 | 0.24 | 0.01 |
| 4F- | 44 | 0.53 | 0.49 | 0.32 | 0.05 |
| 4F | 222 | 0.65 | 0.51 | 0.48 | 0.03 |
| 4F+ | 79 | 0.61 | 0.54 | 0.31 | 0.04 |
| 4F (total) | 345 | 0.62 | 0.51 | 0.43 | 0.02 |
| 1F- | 138 | 1.18 | 0.91 | 1.10 | 0.09 |
| 1F | 553 | 2.10 | 1.33 | 2.13 | 0.09 |
| 1F+ | 296 | 2.93 | 2.22 | 2.63 | 0.15 |
| 1F (total) | 987 | 2.22 | 1.44 | 2.26 | 0.07 |
| P- | 514 | 5.21 | 3.97 | 4.01 | 0.18 |
| P | 1451 | 8.05 | 6.71 | 5.79 | 0.15 |
| P+ | 432 | 12.45 | 11.05 | 8.81 | 0.42 |
| P (total) | 2397 | 8.23 | 6.58 | 6.55 | 0.13 |
| K- | 56 | 16.38 | 16.90 | 6.68 | 0.89 |
| K | 115 | 23.48 | 22.30 | 8.87 | 0.83 |
| K+ | 0 | ~ | ~ | ~ | ~ |
| K (total) | 171 | 21.16 | 20.10 | 8.87 | 0.68 |

**Table 3: Descriptive statistics comparing BHG measurements with the hand hardness test from 68 hand hardness profiles by 33 avalanche technicians.**

To see if there is a difference in the hand hardness indices based on experience, the data from 68 hand hardness profiles was split up based on the avalanche technician's certification. During the 2020/21 and 2021/22 winter field seasons a total of 36 hand hardness profiles by 20 different avalanche technicians with the CAA Operations Level 1 certification were taken with corresponding BHG measurements. During those field seasons a total of 19 hand hardness profiles by 13 different avalanche technicians with the CAA Operations Level 2 certification were taken with corresponding BHG measurements (Fig. 8).






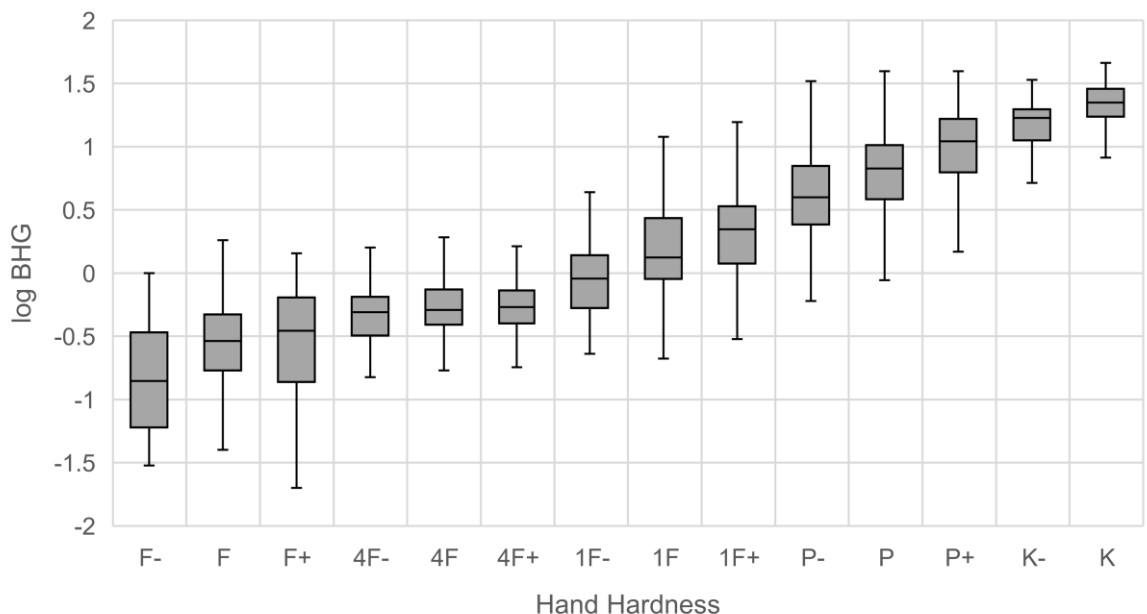

**Figure 7: Log box plot comparing hand hardness indices with BHG measurements from 68 hand hardness profiles by 33 avalanche technicians. The grey boxes represent the 1st and 3rd quartiles, as a measure of spread.  The vertical lines show the full spread of the data.**

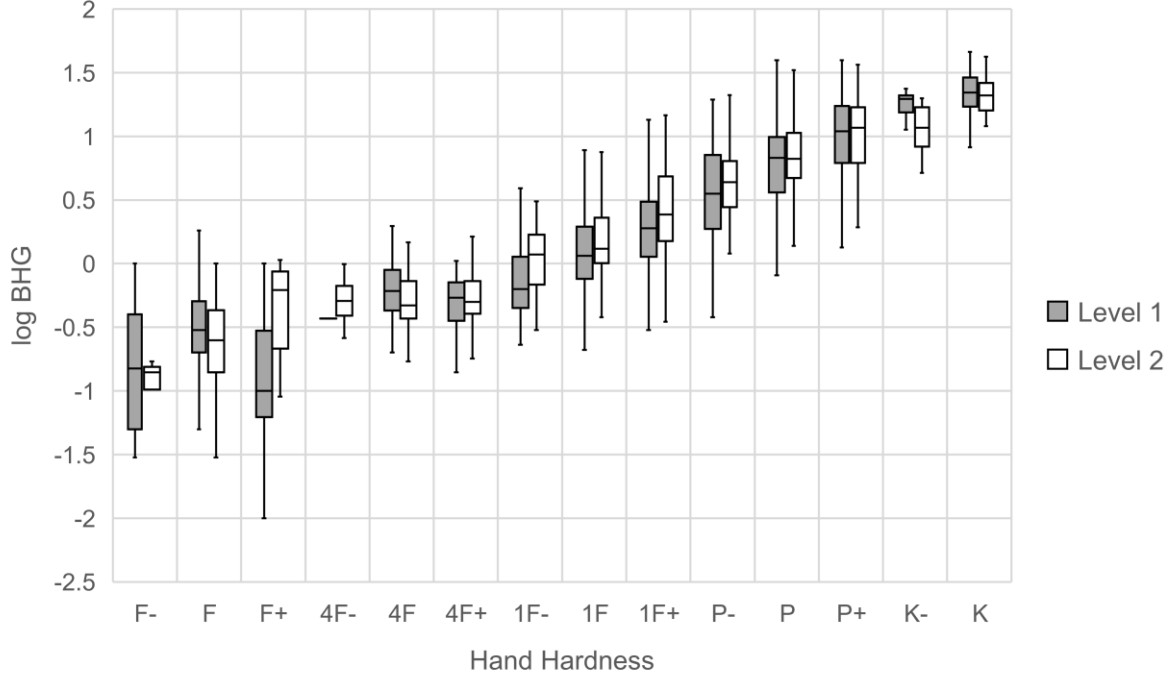

**Figure 8: Log box plot comparing hand hardness indices with BHG measurements by CAA Level 1 and Level 2 operators.**



Log box plots are seen in Fig. 7 and Fig. 8 to visualize the overlap in the data between the hand hardness values. The shaded squares represent the interquartile ranges, the line in the shaded square represents the medians and the whiskers represent the

minimum and maximum ranges found in the log BHG measurements. Comparing the blade hardness with the hand hardness shows significant overlap between the neighbouring hand hardness levels. The log scale of the blade hardness forms an almost linear relationship with the hand hardness as the surface area of each hand hardness level decreases.

Results show there is no significant difference between the F and F+ indices (Mann-Whitney, p = 0.099). As well there is no significant difference between the 4F-, 4F, and 4F+ indices (Kruskal-Wallis, p = 0.235). Level 2 operators showed no statistical

difference between F+, 4F-, 4F, and 4F+ hand hardness indices (Kruskal-Wallis, p = 0.277). All other hand hardness indices have statistically significant differences in their median values (p < 0.01).

Testing the reproducibility of the hand hardness test and the BHG, avalanche technicians took corresponding hand hardness and BHG measurements one after another. Throughout the research a total of 286 snow layers were compared with the hand hardness test and 208 snow layers were compared with the BHG. Results of the reproducibility of the hand hardness test and

the BHG are seen in Fig. 9.

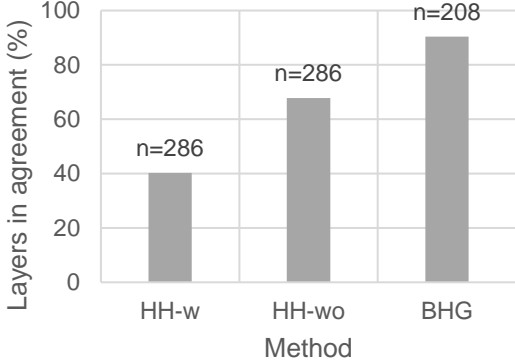

**Figure 9: Side by side snow hardness replication of the hand hardness test with ± indices (HH-w), the hand hardness test without ± indices (HH-wo), and the BHG.**

Figure 9 indicates that the percentage of layers in agreement between avalanche technicians was 90.4% when utilizing the

BHG, in contrast to 40.2% when employing the hand hardness test with the ± indices.

This comparison highlights the superiority of the BHG over the hand hardness test in measuring snow hardness amongst avalanche technicians. Excluding the ± notations, the consistency of the hand hardness test amongst avalanche technicians significantly increased as the percentage of layers in agreement rose to 67.8%.




## 6 Discussion

### 6.1 Insertion Rate

The insertion rate of the BHG is important to consider while using the gauge to measure the hardness of snow. The data supports there is a statistically significant difference between fast ($\approx$ 10 cm/s) and slow ($\approx$ 1-3 cm/s) insertion rates, Wilcoxon signed-rank test ($p < 0.01$), as seen in Figure 3.1. The data also indicated slow insertion rates result in higher data scatter in each snow layer as seen in Figure 3.3. As the gauge is pushed into the snow with a slow insertion rate, it may be the case, that the blade allows compaction of the snow ahead of the blade resulting in the higher and less consistent measurements. Further work on the microscopic effects of the blade would be needed to verify the potential compaction of snow. There is further variability within each measurement, for both the fast and slow insertion rates, as each measurement was taken by the author not a machine.

The original snow resistograph developed by Bradley in 1966 used a thin-blade to measure the hardness of snow. The operator of the snow resistograph pushed the blade attached to a probe down to the bottom of the snowpack, rotated it 90 degrees, and then withdrew the blade upwards with the resistance of the snow being recorded against the thin-blade. The varying rate of withdrawing the blades from the snow resulted in varied hardness measurements. If the rate of withdrawal was too slow there was compaction of snow against the blade. Through multiple trials it was found that the withdrawal rate of 10 cm/s was the optimum rate for consistency (Bradley, 1966). The results Bradley found are similar to the results of this study in relation to the BHG.

In 1977, Fukue used a thin-blade apparatus to measure the hardness of snow. The apparatus included a thin-blade, 12 mm wide and 0.6 mm thick, mounted onto an actuator that mechanically inserted the blade into the snow. During the testing it was discovered there was a ductile to brittle transition at a penetration rate of 0.025 cm/s with the snow exhibiting ductile properties below 0.025 cm/s and the brittle properties above 0.025 cm/s. By measuring the hardness of snow above 0.025 cm/s it can be assumed that the thin-blade penetration was measuring snow characteristics without the change of the initial properties such as density. Increasing the insertion rate to 0.06 cm/s resulted in consistent thin-blade force measurements (Fukue, 1977). Although the current study did not look at the change of ductile to brittle properties by using the fast insertion rate of 10 cm/s it is measuring the snow hardness in the brittle range identified by Fukue.

During the first prototyping of the BHG done by Borstad and McClung, they found no statistically significant difference between fast ($\approx$ 10 cm/s) and slow (1-3 cm/s) were measured subjectively by the operator and done by only one operator to reduce variability. That study used the initial BHG prototype which they concluded had limitations in measuring snow hardness in soft snow, needed a higher a resolution and did not work well in cold temperatures (Borstad & McClung, 2011). The current study used the latest model of the BHG produced by Fraser Instruments Ltd that has more resolution (precise to 0.05 N) and an operating temperature of (-20 to 40°C).

With these results and confirmation with past literature the author recommends using a fast insertion rate of approximately 10 cm/s with the BHG into each snow layer to gain consistent results with the BHG. For the BHG to be used as a standard for





measuring snow hardness it is imperative to use techniques that gain consistency in measurements amongst users. Using the fast insertion rate will result in more consistent measurements amongst all users.

## 6.2 Orientation

In homogenous layers of snow, determined manually by the author, it was found the data supported a statistical difference between using the gauge parallel or perpendicular to the snowpack (Wilcoxon signed-rank test, $p < 0.01$). By using the gauge parallel to the snow, the user can find the differences in hardness throughout each snow layer. Snow layers can range from one to hundreds of millimeters in height. Many variables can affect the snowpack and any given snow layer including aspect, wind, temperature, and ground roughness.

Using the BHG parallel to the snowpack and taking measurements roughly two centimeters vertically apart from one another gives an overall hardness profile of the entire snowpack or the layer in question. For thin persistent weak layers such as layers of buried surface hoar it is only possible to measure the hardness of that weak layer by using the gauge parallel to the snowpack. This is due to the size of the blade. The blade's 0.6 mm thickness is designed to measure thin persistent weak layers.

To have the BHG become a standard tool for measuring snow hardness it would be best practice to have a standard way of

using the tool. From this study, it is recommended using the BHG parallel to the snowpack, taking measurements of the snow top to bottom in two cm increments with extra measurements taken at the location of persistent weak layers if they do not line up with the two cm increments.

## 6.3 Hand Hardness

As the hand hardness test is the current standard for measuring snow hardness in Canada, this research set out to further

correlate the BHG with the hand hardness test. The data from the 68 hand hardness profiles with correlating blade hard measurements, seen in Figure 5, resulted in no difference between the ± indices in the four fingers category. The data from the 19 hand hardness profiles from CAA Level 2 Operators, seen in Figure 6, resulted in no difference between F+, 4F-, 4F and 4F+ indices. This illustrates that avalanche technicians have a hard time distinguishing hardness difference in soft snow and that the ± indices do not have meaning in soft snow. These results are similar to what Pogue et al. found in 2018 (Pogue

et al., 2018).

Comparing the replication of the hand hardness test and BHG amongst users resulted in 90.4% of layers agreeing with the BHG while only 40.2% of layers agree within current measurement precision with the hand hardness test amongst avalanche technicians. This supports thatBHG measurements are more consistent amongst users than the hand hardness test.

Data from this research was used to create a blade hardness to hand hardness scale as seen in Table 4. The BHG can be used

as a teaching tool to introduce and improve consistency of the hand hardness test amongst users. It offers the users the ability to feel what 10-15 N of force feels like, which is the insertion force of the hand hardness test. By measuring the blade hardness, the user can identify the corresponding hand hardness by utilizing the provided blade hardness to hand hardness scale. Continuous calibration of the hand hardness test with the BHG could lead to greater consistency amongst avalanche technicians.





| Hand Hardness Index | Blade Hardness (N) |
|---|---|
| Fist | 0 - 0.4 |
| Four Fingers | 0.4 - 1 |
| One Finger | 1 - 4 |
| Pencil | 4 - 14 |
| Knife | 14 - 45 |

**Table 4: Hand Hardness and Blade Hardness Scale.**

The outcomes of this research suggest that the removal of the ± indices from the hand hardness test would enhance the reproducibility among avalanche technicians. However, within a single snow profile, a particular avalanche technician can utilize the ± indices to assess hardness discrepancies between snow layers in each snow profile. Those ± indices will overlap the other indices over time and will not necessarily be reproduced by another technician.

### 6.4 Conclusions

By measuring the snow hardness, one gains knowledge with respect to how well the snow grains are bonding, bending, rupturing, and compacting with one another. This is important as the bonding of snow grains (not density) is the critical factor in determining how snow responds to applied loads and stresses (Shapiro et al., 1997). The International and Canadian standards utilize the hand hardness test for measuring snow hardness (Canadian Avalanche Association, 2016b; Fierz et al., 2009). The hand hardness test, while useful, does not provide precise quantitative snow hardness measurements to the

avalanche industry. The research shows the BHG to be a reliable tool, compact to travel with, easy to use, and produces consistent measurements when measuring snow hardness. For a more accurate and consistent assessment of snow layer hardness over time, it is recommended to measure the snow hardness with the BHG instead of the hand hardness test.

**Author Contribution**

PB and MP designed the experiments and PB carried them out. PB prepared the manuscript with contributions from MP.

**Competing Interests**

The authors declare that they have no conflict of interest.



**Special Issue Statement**

Latest developments in snow science and avalanche risk management research – merging theory and practice.

**Acknowledgements**

This research was made possible by the supervision and guidance provided by Dr. Mark Paetkau with further guidance from Dr. Iain Stewart-Pattterson and Dr. Richard Taylor.

For supporting research and providing access to the primary field research locations, I would like to thank Kyle Hale, Ryan Harvey, Chris Granter, Adam Sheriff, Steve Crowe, Sean Nyilassy, Lisa Roddick, the entire Kicking Horse Mountain Safety team and Kicking Horse Mountain Resort.

Many thanks to the Avalanche Canada Foundation for providing financial support for this research.

For supplying the blade hardness gauges, I would like to thank Fraser Pogue and Grant Statham.

Thankful to Anton Horvath and Whistler Blackcomb for support to carrying out research in their ski area tenure.

Thanks to Brucejack Mountain Safety for continued support for sharing and pursuing research in the avalanche community.

For great discussions of snow and avalanches, I would like to thank Steve Conger and Dr. David McClung.

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
