# Peer review of "Blade Hardness Gauge: Snow Hardness Measuring and Analysis Techniques"

_EGUsphere, 2024_

## Referee Comment (RC1)

Review on "Blade Hardness Gauge: Snow Hardness Measuring and Analysis Techniques"

Review by Melin Walet

General comments

The authors of the manuscript "Blade Hardness Gauge: Snow Hardness Measuring and Analysis Techniques" aimed to test the reliability and integrity of the Blade Hardness Gauge (BHG). In their work, refinements to standard data collection methods and techniques are proposed.

Avalanche safety is a very important topics for infrastructure, backcountry-skiers and ski resorts. To assess the stability of the snowpack, one needs to gain information on the bonding of the snowpack. This is done by measuring the snow hardness. The attempt of the authors to enable a quantitative, relatively easy, affordable way for individuals to measure snow hardness is therefore very relevant.

In general, the manuscript is fairly well-written. Some minor changes are needed for improved readability and easier understanding, both in the grammar as well as when referring to specific figures. The procedure in the measurements is explained in a clear way. Yet, the storyline could be finetuned when it comes to describing the performed tests. The state-of-the-art is described extensively in the manuscript. Caution is needed to not mention irrelevant previous research or to repeat statements throughout the paper.

I therefore recommend changes to the manuscript as specified below before considering publication.

Specific comments

-r30: *"with no standard method of quantitatively measuring snow hardness…"*: this is your research gap. For the reader, this is stated quite briefly and bluntly. I suggest to make it a bit more explicit and perhaps add an additional sentence about this research gap.

-r31: *"to test this":* I would prefer to read something like "in order to achieve this"

-r34-36: is it needed to specify the details of the BHGs directly here in the introduction? Wouldn't it be better to keep the introduction slightly more global and leave the specifications for the Methods chapter?

-r40-43: what is the intention of writing this here? You also repeat this statement at the end of 2.2 and 6.3. Out of these three locations, the introduction seems the least appropriate to me.

-r67 and r125-135: You describe the SMP quite extensively in your manuscript. Perhaps there could be more emphasis on the fact that it is expensive (r67) and that therefore BHG is much more practical for individual usage.

-r104: what is the difference between second and first iteration of the BHG?

-r191: why is it relevant to distinguish what the avalanche technician did and what the researcher did? Is there a difference which the reader should know?

-in the Results chapter, I would appreciate a table/overview of all the experiments which were done. The current description of the various experiments is a bit lengthy and not so clear/concise.

-r221: "pairs of HBG measurements": can you specify what these pairs are?

-r225-226: How does Figure 5 show that there is a statistically significant difference? Should the reader understand this from the Figure? Why do you refer to figure 5 here?

-Figure 5: the x-axis of this figure is not easy to read. Is there another way to design this figure? What should the reader understand from the figure?

-do you mention somewhere the snow profile and/or the snow height where you do the tests? What are the conditions? Can you say anything about the layering of the snow? Would that be relevant for the test results?

-r257: can you explain why it is relevant to distinguish between Level 1 and Level 2 certification?

-Figure 7: is Figure 7 needed as there is also Figure 8? The size of the figures is way better than for example figure 5, which is too small.

-Table 4: I suggest to put table 4 in the results section, as it seems more appropriate there.

Technical corrections

-r56/57: grammatically not correct. You might want to say *"the advantage was that it outputted a graphical hardness profile but **it** had a high cost and high mass*. (add "it"). Otherwise it is part of the advantage.

-throughout the manuscript: you use "unable to" (mostly on page 3) and "utilizing" quite often. Try to vary a bit your wording.

-r84: replace utilize (already used a lot)

-r85: *"with four goals in mind: a) involved a simple technique b)…"* I don't understand the structure/grammar here. Maybe change to *"…with four goals in mind: the technique should a) involve a simple technique, b) minimize…"* etc.

-r87: can you actually say "the testing found…"?

-r134: why a comma after SMP?

-some sentences are standing a bit lonely such as the r135 and r136. Can you try to process them more into the rest of the text?

-in general, more connecting words are missing. It now is a summation of different facts and not really a fascinating story to read. Try to make the text a bit more attractive.

-r149: applying, using, are all synonyms of utilizing. Try to not repeat the same word over and over.

R161: *"Further observations":* it seems more appropriate to say "all other observation".

-r239: *"With the fast insertion rate…":* can you merge this into a full sentence?

-Table 3: It might be useful to add that the columns with Newton as unit, refer to the BHG measurements. Currently this is not obvious from the table alone, i.e. it is not self-explanatory

-r.286-288: I propose to switch the order of the statements. Currently, you first mention that the BHG is superior over the hand hardness test (an important conclusion), and then you seem to weaken the statement by saying that the difference is less pronounced between BHG and the hand hardness test without +- indices. If you want to mostly highlight that the BHG is superior, I would change the formulation and finish the paragraph with mentioning the superiority.

r.293: *"as seen in Figure 3.1":* I think you mean another figure? Which figure do you mean?

r.294. *"in Figure 3.3":* do you mean figure 6?

-r298: *"by the author not a machine?"* try to rephrase this wording to something more professional.

-r304: I suggest to swap the order? Something like *"The results of this study are in agreement with the results from Bradley?"*

-r309: *"and the brittle properties":* I don't think "the" should be written here.

-r314-319: you repeat this from section 2.2. Please decide where you mention it.

-r320 "these results": if you refer to the results of the current study, please state this explicitly to distinguish from the literature results.

-r325: *"it was found.."* add *"that"?*

-r341: "*seen in Figure 5*:" should be figure 7?

-r342: "*seen in Figure 6*:" should be figure 8?

-r343: difference or differences?

-r348: "*that* (add white space!) *BHG"*